

# Geometric expansion of fluctuations and average shadows

Clément Berthière[1], Benoit Estienne[2], Jean-Marie Stéphan[3,4]
and William Witczak-Krempa[5,6,7]

**1** Laboratoire de Physique Théorique, CNRS, Université de Toulouse, France
**2** Sorbonne Université, CNRS, Laboratoire de Physique Théorique et Hautes Energies,
LPTHE, F-75005 Paris, France
**3** Université Claude Bernard Lyon 1, ICJ UMR5208, CNRS, 69622 Villeurbanne, France
**4** ENS de Lyon, CNRS, Laboratoire de Physique, F-69342 Lyon, France
**5** Dèpartement de Physique, Université de Montréal, Montréal, QC, H3C 3J7, Canada
**6** Centre de Recherches Mathématiques, Université de Montréal,
Montréal, QC, H3C 3J7, Canada
**7** Institut Courtois, Université de Montréal, Montréal, QC, H2V 0B3, Canada

## Abstract

Fluctuations of observables provide unique insights into the nature of physical systems, and their study stands as a cornerstone of both theoretical and experimental science. Generalized fluctuations, or cumulants, provide information beyond the mean and variance of an observable. In this paper, we develop a systematic method to determine the asymptotic behavior of cumulants of local observables as the region becomes large. Our analysis reveals that the expansion is closely tied to the geometric characteristics of the region and its boundary, with coefficients given by convex moments of the connected correlation function: the latter is integrated against intrinsic volumes of convex polytopes built from the coordinates, which can be interpreted as average shadows. A particular application of our method shows that, in two dimensions, the leading behavior of odd cumulants of conserved quantities is topological, specifically depending on the Euler characteristic of the region. We illustrate these results with the paradigmatic strongly-interacting system of two-dimensional quantum Hall state at filling fraction 1/2, by performing Monte-Carlo calculations of the skewness (third cumulant) of particle number in the Laughlin state.

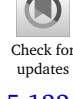
doi:10.21468/SciPostPhys.19.5.122

# 1 Introduction

Understanding, describing, and quantifying the behavior of physical systems hinges on the ability to predict and measure physically meaningful quantities. One particularly effective procedure involves selecting a subsystem, typically by choosing a simple spatial region, and analyzing the scaling behavior of certain observables as the subsystem becomes large. This approach has proven highly successful across various physical systems and with different types of probes, allowing for the extraction of valuable information such as insights into long-range correlations, quantum criticality, and topological properties. The choice of region is sometimes dictated by the experimental setup, and other times by convenience. This choice, however, can significantly impact the physics being revealed. Disentangling physical information from the geometric characteristics of the selected region represents a complex challenge, often addressed on a case-by-case basis, depending on the geometry and specific theory in question (e.g., [1–19]). This underscores the need for general, theory-independent results in this area. In this paper, we solve this problem to the leading orders for *the cumulants of any local observable, for arbitrary smooth regions in any translation invariant theories,* under physically natural locality assumptions.

The $m$–th bipartite cumulant of a local observable in a finite region $A \subset \mathbb{R}^d$ (see Fig. 1(a)) may be written as

$$C_m(A) = \int_{A^m} d\mathbf{r}_1 \dots d\mathbf{r}_m \langle \rho(\mathbf{r}_1) \dots \rho(\mathbf{r}_m) \rangle_c \,, \tag{1}$$

with $\rho(\mathbf{r})$ the local density associated to the observable and $\langle \rho(\mathbf{r}_1) \dots \rho(\mathbf{r}_m) \rangle_c$ its connected $m$–point function. This correlation function is assumed not to suffer from UV singularities at coincident points, except for Dirac-delta singularities which typically occur when counting particles (full-counting statistics). We assume the connected correlation function is translation invariant, and decays sufficiently fast whenever any of its argument goes to infinity, see Sec. 3.

We are interested in the asymptotic expansion of cumulants for large spatial regions. To investigate this, we dilate region $A$ by a factor $\lambda$. In the limit of large $\lambda$, it is believed that the $m$-th cumulant admits an expansion of the form

$$C_m(\lambda A) = c_0 \lambda^d + c_1 \lambda^{d-1} + c_2 \lambda^{d-2} + \cdots \,, \tag{2}$$

where the coefficients $c_i$ depend on the connected $m$-point correlation function, the space dimension $d$, and the geometry of $A$. This type of expansion has been implicitly assumed in earlier works such as [19] and [20], where the $\lambda^0$ contribution to the variance and higher cumulants, respectively, was studied in two-dimensional systems. Similar expansions have also been conjectured for entanglement entropies in systems with sufficiently local correlations [21].

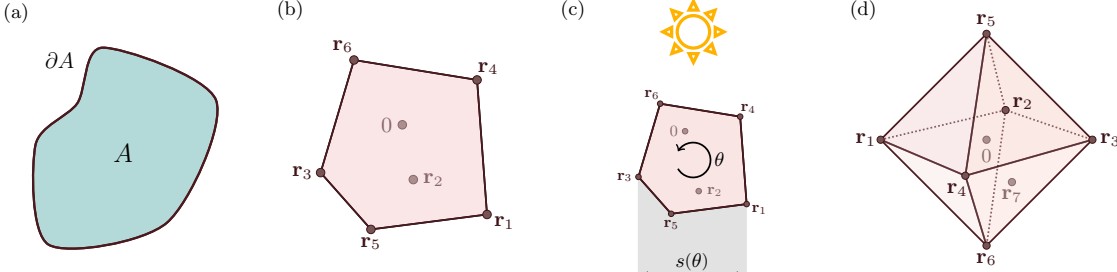

Figure 1: (a) A large subregion $A$ in $\mathbb{R}^2$ with a smooth boundary $\partial A$. (b) Convex hull $H = H(\mathbf{r}_1, \ldots, \mathbf{r}_{m-1}, 0)$ relevant to the expansion of $C_m(A)$ in $d = 2$ dimensions (an example with $m = 7$ is shown). In this case, $2\pi v_1 = \mathrm{vol}(\partial H)$ is the perimeter of the hull, see (9), while $2\pi v_2 = \mathrm{vol}(H)$ is the area of the hull, see (10). (c) Mean shadow interpretation of $v_1$. Here $s(\theta)$ is the length of the shadow projected onto a fixed straight line for some orientation of $H$ labeled by the angle $\theta$ (the fixed light source is far away in a direction perpendicular to the line). Averaging $s(\theta)$ over all angles yields the mean width, which is twice $v_1$. (d) Convex hull $H$ relevant to the computation of $C_m(A)$ in $d = 3$ dimensions (an example with $m = 8$ is shown). In this case, $v_1$ is given by (9) and involves the exterior dihedral angles between the two faces adjacent to the edges, while $8\pi v_2 = \mathrm{vol}(\partial H)$ (see (10)) is the surface area of the hull, i.e. the sum of the surface area of all faces.

In this work, we rigorously establish the validity of such an expansion in broad generality and derive explicit expressions for the coefficients $c_i$. Throughout, we assume that the underlying theory is isotropic and translation-invariant. While isotropy is not required for the validity of our approach, it simplifies the structure of the coefficients, which can then be expressed in terms of invariant geometric quantities known as *intrinsic volumes* (see Appendix).

## 2 Summary of the main results

We develop a method to compute bipartite cumulants of local observables. We establish a general formula for the coefficients $c_0, c_1, c_2$ in the expansion (2) of the cumulants at large regions:

$$c_0 = s_0[f]\,\mathrm{vol}(A)\,, \tag{3}$$

$$c_1 = s_1[f]\,\mathrm{vol}(\partial A)\,, \tag{4}$$

$$c_2 = s_2[f]\int_{\partial A} d\sigma\,\mathrm{tr}\,\kappa\,, \tag{5}$$

where $d\sigma$ is the surface element on the boundary $\partial A$, and $\kappa$ is the extrinsic curvature tensor associated to $\partial A$. First, $c_1$ is proportional to the size, or "area", of the boundary, and is well-known (e.g., [22]) as area-law term. Secondly, $c_2$ is sensitive to the curvature of the boundary. Finally, $s_j[f]$ are integrals that depend on the physical state through the connected correlation functions $f(\mathbf{r}_1, \ldots, \mathbf{r}_{m-1}) = \langle \rho(\mathbf{r}_1) \ldots \rho(\mathbf{r}_{m-1})\rho(0)\rangle_c$, but not on $A$.

A remarkable feature of (3–5) is the complete factorisation between the geometric and physics contributions; this was called *superuniversality* in [19] where the same phenomenon occurred for the variance $C_2$ of regions with corners. The factorisation pattern becomes more complicated for higher-order $c_j$'s; for instance, both $\mathrm{tr}\,\kappa^2$ and $(\mathrm{tr}\,\kappa)^2$ appear in $c_3$, while the number of such terms quickly increases with the order. Our method theoretically enables the explicit determination of all of them, but this will be covered in a separate discussion [23].

Let us now write our main formulas for the $s_j[f]$:

$$s_0[f] = \int d\mathbf{r}\, f(\mathbf{r}_1, \ldots, \mathbf{r}_{m-1}), \tag{6}$$

$$s_1[f] = -\int d\mathbf{r}\, v_1[\mathbf{r}_1, \ldots, \mathbf{r}_{m-1}]\, f(\mathbf{r}_1, \ldots, \mathbf{r}_{m-1}), \tag{7}$$

$$s_2[f] = \int d\mathbf{r}\, v_2[\mathbf{r}_1, \ldots, \mathbf{r}_{m-1}]\, f(\mathbf{r}_1, \ldots, \mathbf{r}_{m-1}), \tag{8}$$

where integrals are taken over $\mathbb{R}^d$, and $d\mathbf{r}$ is a shorthand for $d\mathbf{r}_1 \ldots d\mathbf{r}_{m-1}$. The functions $v_1$ and $v_2$ are obtained by considering the convex hull $H$ generated by all the $\mathbf{r}_j$ and the origin (see Sec. 3). This hull is a convex polytope, which can be determined numerically very efficiently [24]. For this reason, we shall call the $s_j$ "convex moments" of $f$. Examples of such hulls are shown in Fig. 1 (b,d).

Before further discussing convex moments, we mention an important simplification when the observable is conserved. In that case [20] $s_0$ always vanishes, $s_1$ vanishes for odd cumulants, and $s_2$ vanishes for even cumulants.

**Intrinsic volumes and average shadows.** The functions $v_j$ appearing in the convex moments possess a beautiful geometric interpretation. Specifically, $v_1$ is half the mean width, a well-known measure in the context of convex geometry [25, 26]. It can be computed in arbitrary dimension, but the formulas are particularly nice and simple in two and three dimensions:

$$v_1^{2d} = \frac{1}{2\pi}\text{vol}(\partial H), \qquad v_1^{3d} = \frac{1}{8\pi}\sum_{e \in H} \ell_e \varphi_e. \tag{9}$$

In 2d, $v_1$ is proportional to the perimeter of the hull $H$, while in 3d the sum runs over all edges $e$, $\ell_e$ being the length of $e$ and $\varphi_e$ the exterior dihedral angle between the two faces adjacent to $e$. In general, $v_1$ is the unique continuous measure of a convex body which behaves extensively under union of convexes, and has dimension of a length.[1]

Similarly, $v_2$ is the unique such measure which has dimension of an area. The explicit expressions are

$$v_2^{2d} = \frac{\text{vol}(H)}{2\pi}, \qquad v_2^{3d} = \frac{\text{vol}(\partial H)}{8\pi}. \tag{10}$$

In 2d, $v_2$ is the area of the polygon $H$, and in 3d it is, famously [27–30], the surface area of the polyhedron $H$.

More generally, $v_1$ and $v_2$ are examples of so-called *intrinsic volumes*. In $d$ dimensions, there are $d$ such non-trivial volumes $v_j$. Each can be interpreted as the average $j$–dimensional shadow of $H$, that is the mean volume of the projection onto a $j$–dimensional subspace, averaged over all possible orientations of $H$ (see Fig. 1 (c)). Only $v_1$ and $v_2$ enter the cumulants expansion up to third order, though higher-order corrections are more complicated. The appearance of such intrinsic volumes of $H$ in our problem is remarkable, since the starting point has nothing to do with convexes, the only geometric input being the choice of a smooth non-necessarily convex region $A$.

For concreteness, we display these intrinsic volumes entering the convex moments in 2d for the variance $(C_2)$

$$v_1[\mathbf{r}_1] = \frac{|\mathbf{r}_1|}{\pi}, \qquad v_2[\mathbf{r}_1] = 0, \tag{11}$$

---

[1] The precise statement goes under the name of Hadwiger's theorem.

and the skewness ($C_3$)

$$v_1[\mathbf{r}_1, \mathbf{r}_2] = \frac{|\mathbf{r}_1| + |\mathbf{r}_2| + |\mathbf{r}_1 - \mathbf{r}_2|}{2\pi}, \qquad v_2[\mathbf{r}_1, \mathbf{r}_2] = \frac{|\mathbf{r}_1 \wedge \mathbf{r}_2|}{4\pi}. \tag{12}$$

Those expressions generalize to any dimension, as we show in Sec. 3 (e.g., $v_1[\mathbf{r}_1] = \frac{\mathrm{vol}(\mathbb{S}^{d-2})}{(d-1)\mathrm{vol}(\mathbb{S}^{d-1})}|\mathbf{r}_1|$ where $\mathbb{S}^{d-1}$ is the unit sphere in $\mathbb{R}^d$). By convention and for continuity reasons in the limit of flat triangles, the perimeter of a segment is twice its length.

**The case of two dimensions.** As already mentioned, the expressions for $s_1$ and $s_2$ greatly simplify in two dimensions. Furthermore, we may use the Gauss–Bonnet formula $\int_{\partial A} d\sigma \, \kappa = 2\pi \chi_A$ which relates the boundary curvature to the Euler characteristic $\chi_A$ of region $A$. Expression (5) then simplifies to $c_2 = s_2[f] 2\pi \chi_A$. This result is especially relevant for odd cumulants if the observable under consideration is conserved. In that case, both the volume term and the area term vanish, which means odd cumulants converge to the constant

$$C_{2n+1}(A) = s_2[f] 2\pi \chi_A. \tag{13}$$

The Euler characteristic is a topological invariant, hence for a given theory, odd cumulants remain unaffected by smooth deformations of the region $A$. This is particularly interesting since most studies of cumulants focus on the highly symmetric disk geometry [31, 32].

**Relation to corner terms.** So far, we have only discussed smooth geometries. Polygonal domains, for example, present sharp corners which affect dramatically the behavior of cumulants: the area-law term remains the same, but all cumulants now have a constant piece, which can be quite complicated [20].

Each corner with opening angle $\theta$ (see Fig. 2 on the left) contributes additively a term $a_m(\theta)$ to this constant, with $a_2(\theta)$ being the only term fully known and explicitly described [19]. It is natural to ask whether one can reconstruct the smooth constant piece in the cumulants by an Archimedean trick, which is depicted in Fig. 2. To facilitate comparison with existing literature, we focus on the case where the observable is conserved.

Consider a regular polygon $A_N$ with $N$ vertices, each with interior angle $\theta_N = \pi(N-2)/N$ (see Fig. 2 on the right). The associated corner term in the cumulants is known to vanish in the limit $\theta \to \pi$. For odd cumulants of conserved observables, the precise behavior of the corner term is $a_m(\theta) \sim \sigma_m(\pi - \theta)$, with some a priori unknown coefficient $\sigma_m$. This translates into the following relation

$$\lim_{N \to \infty} N a_m(\theta_N) = 2\pi \sigma_m. \tag{14}$$

Since the regular polygon $A_N$ becomes a disk in this limit, it is reasonable to expect (14) to match the smooth result (13). By this reasoning, we are able to use (5) to predict a highly nontrivial exact formula for $\sigma_m$,

$$\sigma_m = s_2[f], \tag{15}$$

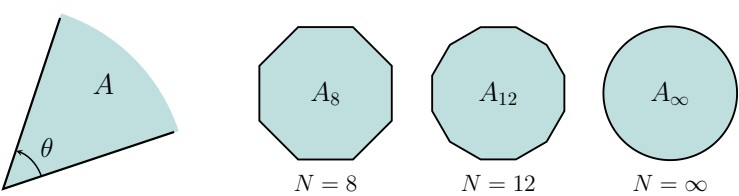

Figure 2: Left: a corner with angle $\theta$ contributes a constant term $a_m(\theta)$ to all cumulants. Right: sequence of convex regular polygons $A_N$, with interior angles $\theta_N = \pi(N-2)/N$ and fixed perimeter, the $N \to \infty$ limit of which is a disk.

where $s_2[f]$ is given by (8). We expect this procedure to work generally; an example where the constants $\sigma_m$ were only known numerically is discussed in Sec. 4. This finding illustrates the fact that corner terms of odd cumulants are very rich, since the result for smooth regions is implicitly contained in the linear behavior near $\theta = \pi$. Another qualitative difference is that corner terms are neither topological nor superuniversal in general [20].

Finally, it is worth noting that this procedure also applies to even cumulants, though it yields less information. Indeed, the corresponding corner terms vanish quadratically [20], hence performing the limit yields a vanishing constant term consistent with our general smooth result $c_2 = 0$ for conserved observables.

## 3 The volume method

### 3.1 Physical assumptions

We design a method to study the asymptotics of cumulants for arbitrary smooth large compact regions $A \subset \mathbb{R}^{d \geq 2}$,

$$C_m(A) = \int_{A^m} d\mathbf{r}_1 \ldots d\mathbf{r}_m \, \langle \rho(\mathbf{r}_1) \ldots \rho(\mathbf{r}_m) \rangle_c \,, \tag{16}$$

where $\langle \ldots \rangle_c$ denotes the usual connected correlation functions. Before performing this expansion, let us state our main physical assumptions, and how they immediately translate to the correlation functions.

- *Translation invariance*: $\langle \rho(\mathbf{r}_1 + \mathbf{s}) \ldots \rho(\mathbf{r}_m + \mathbf{s}) \rangle_c = \langle \rho(\mathbf{r}_1) \ldots \rho(\mathbf{r}_m) \rangle_c$ for any $\mathbf{s} \in \mathbb{R}^d$. Thus all information about correlations is encoded in $f(\mathbf{r}_1, \ldots, \mathbf{r}_{m-1}) = \langle \rho(\mathbf{r}_1) \ldots \rho(\mathbf{r}_{m-1}) \rho(0) \rangle_c$.

- *Rotational invariance*: $\langle \rho(\mathbf{r}'_1) \ldots \rho(\mathbf{r}'_m) \rangle_c = \langle \rho(\mathbf{r}_1) \ldots \rho(\mathbf{r}_m) \rangle_c$ where $\mathbf{r}'_j = \mathcal{R}(\mathbf{r}_j)$ for $\mathcal{R}$ any rotation in $\mathbb{R}^d$.

- *Locality*. This implies that $f(\mathbf{r}_1, \ldots, \mathbf{r}_{m-1})$ decays sufficiently fast whenever any argument $\mathbf{r}_j$ goes to infinity. This includes the case where a subset of variables go to infinity while staying close to each other. To guarantee an expansion of cumulants at all orders, $f$ should in principle decay faster than any power law. However, if we truncate the expansion at order $\lambda^{d-j}$, a power law decay with exponent greater than $d + j$ is sufficient. The function $f$ is assumed to be continuous, but our main result nevertheless allows for $\delta$-function singularities at coincident points, which are relevant in the context of counting statistics.

### 3.2 Volume and isotropic volume

Let us exploit our three assumptions in succession, starting with translation invariance first. We have

$$C_m(A) = \int d\mathbf{r}_1 \ldots d\mathbf{r}_m \, 1_A(\mathbf{r}_1) \ldots 1_A(\mathbf{r}_m) \, \langle \rho(\mathbf{r}_1) \ldots \rho(\mathbf{r}_m) \rangle_c \,, \tag{17}$$

where $1_A(\mathbf{r}) = 1$ if $\mathbf{r} \in A$ and $1_A(\mathbf{r}) = 0$ otherwise. A change of variable yields

$$C_m(A) = \int d\mathbf{r}_1 \ldots d\mathbf{r}_{m-1} \left( \int d\mathbf{r} \, 1_A(\mathbf{r}) 1_A(\mathbf{r} + \mathbf{r}_1) \ldots 1_A(\mathbf{r}_{m-1} + \mathbf{r}) \right) f(\mathbf{r}_1, \ldots, \mathbf{r}_{m-1})$$

$$= \int d\mathbf{r}_1 \ldots d\mathbf{r}_{m-1} \, \mathcal{G}_A(\mathbf{r}_1, \ldots, \mathbf{r}_{m-1}) f(\mathbf{r}_1, \ldots, \mathbf{r}_{m-1}), \tag{18}$$

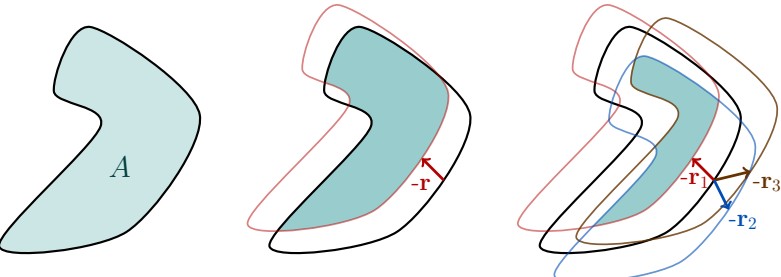

Figure 3: Illustration of the volume $\mathcal{G}_A$ in two dimensions. *Left*: region $A$ with interior shown in light green, and smooth boundary. *Center*: volume $\mathcal{G}_A(\mathbf{r}) = \text{vol}[A \cap (A-\mathbf{r})]$ shown in green, for the choice of vector $\mathbf{r}$ (red arrow). *Right*: volume $\mathcal{G}_A(\mathbf{r}_1, \mathbf{r}_2, \mathbf{r}_3) = \text{vol}[A \cap (A-\mathbf{r}_1) \cap (A-\mathbf{r}_2) \cap (A-\mathbf{r}_3)]$ in green, for the choice of vectors $\mathbf{r}_1, \mathbf{r}_2, \mathbf{r}_3$ (red, blue, brown arrows).

where

$$\mathcal{G}_A(\mathbf{r}_1, \ldots, \mathbf{r}_{m-1}) = \text{vol}\left[A \cap (A-\mathbf{r}_1) \cap \ldots \cap (A-\mathbf{r}_{m-1})\right], \tag{19}$$

is a volume which depends only on region $A$ and the set of vectors $\mathbf{r}_1, \ldots, \mathbf{r}_{m-1}$, see Fig. 3 above for an illustration. This purely geometric quantity has been considered in the context of Fredholm determinants [33–37], and was also exploited to study corner terms in two dimensions [20]. Relevant to the variance, $\mathcal{G}_A(\mathbf{r})$ is known as covariogram or geometric covariogram [25, 38] in the context of geostatistics, set covariance function in mathematical morphology; $\mathcal{G}_A(\mathbf{r})/\text{vol}(A)$ is called the autocorrelation function in the context of small angle X-ray scattering [39].

Second, we may exploit the invariance of $f$ under simultaneous rotation of its arguments $\mathbf{r}_j$, to rewrite (18) as

$$C_m(A) = \int d\mathbf{r}_1 \ldots d\mathbf{r}_{m-1} \overline{\mathcal{G}_A}(\mathbf{r}_1, \ldots, \mathbf{r}_{m-1}) f(\mathbf{r}_1, \ldots, \mathbf{r}_{m-1}), \tag{20}$$

with $\overline{\mathcal{G}_A}$ the rotation-averaged volume, or isotropic volume

$$\overline{\mathcal{G}_A}(\mathbf{r}_1, \ldots, \mathbf{r}_{m-1}) = \int_{\mathcal{R} \in SO(d)} d\mu(\mathcal{R}) \, \mathcal{G}_A\big(\mathcal{R}(\mathbf{r}_1), \ldots, \mathcal{R}(\mathbf{r}_{m-1})\big). \tag{21}$$

Here the integration is with respect to the Haar measure in the group of rotations $SO(d)$ (equivalently, one can average over rotations of $A$ instead of the $\mathbf{r}_j$). See Fig. 4 for an example.

We are interested in the expansion of $C_m(\lambda A)$ for large $\lambda$. Using finally our locality assumption, this boils down to finding an expansion of $\mathcal{G}_{\lambda A}(\mathbf{r}_1, \ldots, \mathbf{r}_{m-1})$ for large $\lambda$ and fixed set of vectors $\mathbf{r}_1, \ldots, \mathbf{r}_{m-1}$. Since

$$\mathcal{G}_{\lambda A}(\mathbf{r}_1, \ldots, \mathbf{r}_{m-1}) = \lambda^d \mathcal{G}_A(\mathbf{r}_1/\lambda, \ldots, \mathbf{r}_{m-1}/\lambda), \tag{22}$$

it suffices to find an expansion of $\mathcal{G}_A$ for small displacements $\mathbf{r}_j$, which can be done in principle to any order using tools coming from differential geometry [36, 37]. Such an expansion can then be used to access an expansion of $\overline{\mathcal{G}_A}$ for small arguments. This is what we do in the next subsection.

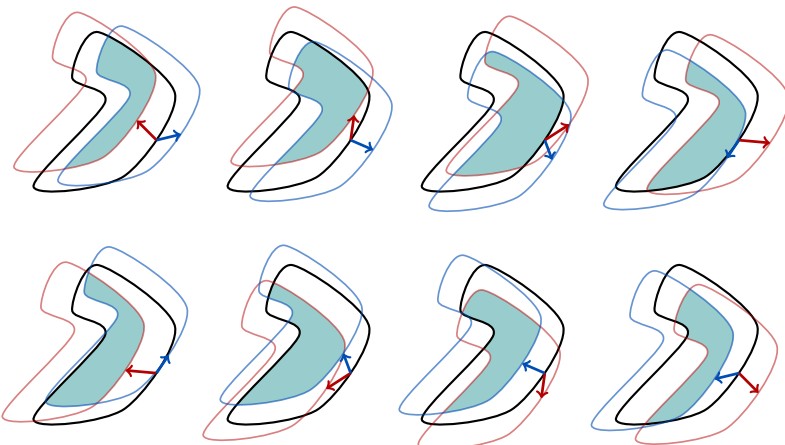

Figure 4: Illustration of the isotropic volume $\overline{\mathcal{G}_A}(\mathbf{r}_1, \mathbf{r}_2)$ in two dimensions. The choice of region $A$ is the same as in the previous figure, the vectors $\mathbf{r}_1, \mathbf{r}_2$ are shown top left in red and blue. The translated regions $A - \mathbf{r}_1$ and $A - \mathbf{r}_2$ are shown in lighter colors to better identify the initial region $A$. The volume $\mathcal{G}_A(\mathbf{r}_1, \mathbf{r}_2)$ is represented in green. From top left to top right and bottom right to bottom left: $\mathcal{G}_A(\mathbf{r}_1^\theta, \mathbf{r}_2^\theta)$ where the $\mathbf{r}_j^\theta$ are obtained from the $\mathbf{r}_j$ by a rotation of angle $\theta$, with successive clockwise increments by $\pi/4$ shown. $\overline{\mathcal{G}_A}(\mathbf{r}_1, \mathbf{r}_2)$ is obtained by averaging $\mathcal{G}_A(\mathbf{r}_1^\theta, \mathbf{r}_2^\theta)$ over all angles $\theta$. Vectors $\mathbf{r}_1$ and $\mathbf{r}_2$ were chosen to be not so small to magnify curvature effects.

### 3.3 Derivation of the main volume expansion

The first step is to rewrite $\mathcal{G}_A$ as

$$\mathcal{G}_A(\mathbf{r}_1, \ldots, \mathbf{r}_{m-1}) = \text{vol} A - \mathcal{V}_A(\mathbf{r}_1, \ldots, \mathbf{r}_{m-1}), \tag{23}$$

with ($B$ is the complement of $A$ in $\mathbb{R}^d$)

$$\mathcal{V}_A(\mathbf{r}_1, \ldots, \mathbf{r}_{m-1}) = \text{vol}\left(A \cap [(B - \mathbf{r}_1) \cup \ldots \cup (B - \mathbf{r}_m)]\right). \tag{24}$$

The volume $\mathcal{V}_A$ is shown in light green in Fig. 5. For small $\mathbf{r}_j$ this volume can be evaluated by localizing to the boundary $\partial A$, which we assume to be smooth. More precisely, we have the small $\epsilon$ expansion [34, 35],

$$
\begin{aligned}
\mathcal{V}_A(\epsilon \mathbf{r}_1, \ldots, \epsilon \mathbf{r}_{m-1}) = \int_{\partial A} d\sigma \max_{1 \leq j \leq m-1}\left(0, \epsilon \mathbf{r}_j.\mathbf{n} + \frac{\epsilon^2}{2}\kappa_{ab} r_j^a r_j^b\right) \\
- \frac{\epsilon^2}{2}\int_{\partial A} d\sigma \kappa_a^a \max_{1 \leq j \leq m-1}\left(0, \mathbf{r}_j.\mathbf{n}\right)^2 + o(\epsilon^2),
\end{aligned}
\tag{25}
$$

where $d\sigma = d\sigma(\mathbf{x})$ denotes the surface element at point $\mathbf{x}$ on the boundary $\partial A$, $\kappa = \kappa_{ab}$ is the associated curvature tensor (second fundamental form), and $\kappa_a^a = \text{tr}\,\kappa$. We use the shorthand notation $\max_{1 \leq j \leq m-1}(0, x_j)$ for $\max(0, x_1, \ldots, x_{m-1})$, as well as Einstein's summation conventions. Each vector $\mathbf{r}_j$ may be written as $\mathbf{r}_j = r_j^0 \mathbf{n} + \mathbf{r}_j^\top$ where $\mathbf{n} = \mathbf{n}(\mathbf{x})$ is the unit outer normal at a point $\mathbf{x}$ of the boundary, and $\mathbf{r}_j^\top$ denotes the tangential component of $\mathbf{r}_j$. We may write $\mathbf{r}_j^\top = \sum_{a=1}^{d-1} r_j^a \mathbf{e}_a$, where $\{\mathbf{e}_a\}$ forms a basis of the tangent space at $\mathbf{x}$. Similar to $\overline{\mathcal{G}_A}$, one can define

$$\overline{\mathcal{V}_A}(\mathbf{r}_1, \ldots, \mathbf{r}_{m-1}) = \int_{\mathcal{R} \in SO(d)} d\mu(\mathcal{R})\, \mathcal{V}_A(\mathcal{R}(\mathbf{r}_1), \ldots, \mathcal{R}(\mathbf{r}_{m-1})). \tag{26}$$

Taking average over rotations yields nice simplification. To better understand these, we study the first-order term in the next subsection, before taking on the next one.

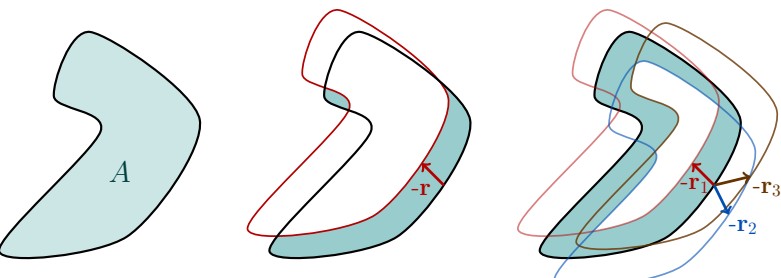

Figure 5: Exact same setup as in Fig. 3, but for $\mathcal{V}_A$ instead of $\mathcal{G}_A$. *Left*: region $A$. *Middle*: volume $\mathcal{V}_A(\mathbf{r})$ of $A \cap (B - \mathbf{r}_1)$ shown in green. *Right*: volume $\mathcal{V}_A(\mathbf{r}_1, \mathbf{r}_2, \mathbf{r}_3)$ of $A \cap [(B - \mathbf{r}_1) \cup (B - \mathbf{r}_2) \cap (B - \mathbf{r}_3)]$ (green) used to compute the fourth cumulant $C_4(A)$ in two dimensions. For small $\mathbf{r}_j$ most of the volume is localized near the boundary $\partial A$.

### 3.3.1 First order contribution

Start with

$$
\begin{aligned}
\overline{\mathcal{V}_A}(\mathbf{r}_1, \ldots, \mathbf{r}_{m-1}) &= \int_{\mathcal{R} \in SO(d)} d\mu(\mathcal{R}) \int_{\partial A} d\sigma \max_{1 \leq j \leq m-1} \left(0, \epsilon \, \mathcal{R}(\mathbf{r}_j).\mathbf{n}\right) + O(\epsilon^2) \\
&= \epsilon \int_{\partial A} d\sigma \int_{\mathbf{u} \in \mathbb{S}^{d-1}} d\nu(\mathbf{u}) \max_{1 \leq j \leq m-1}(0, \mathbf{r}_j.\mathbf{u}) + O(\epsilon^2) \\
&= \epsilon \, \mathrm{vol}(\partial A) \, v_1[\mathbf{r}_1, \ldots, \mathbf{r}_{m-1}] + O(\epsilon^2),
\end{aligned}
\tag{27}
$$

where we have replaced the average over rotations by a unit sphere average ($d\nu(\mathbf{u})$ denotes the uniform measure on the unit sphere) and used the fact that the resulting integral over $\partial A$ does not depend any more on the orientation of the unit vector $\mathbf{n}$. We define $v_1$ as

$$
v_1[\mathbf{r}_1, \ldots, \mathbf{r}_{m-1}] = \int_{\mathbf{u} \in \mathbb{S}^{d-1}} d\nu(\mathbf{u}) \max_j(0, \mathbf{r}_j.\mathbf{u}),
\tag{28}
$$

which is a known quantity in the context of integral (convex) geometry, see Appendix. To explain why that is, let us recall the convex Hull $H = H(\mathbf{r}_1, \ldots, \mathbf{r}_{m-1}, 0)$. It is easy to check that

$$
\max_{1 \leq j \leq m-1}(0, \mathbf{r}_j.\mathbf{u}) = \max_{\mathbf{r} \in H}(\mathbf{r}.\mathbf{u}).
\tag{29}
$$

In words, for a convex polytope, the max is attained for $\mathbf{r}$ being one of the vertices. This means

$$
v_1[\mathbf{r}_1, \ldots, \mathbf{r}_{m-1}] = \int_{\mathbf{u} \in \mathbb{S}^{d-1}} d\nu(\mathbf{u}) \max_{\mathbf{r} \in H}(\mathbf{r}.\mathbf{u}),
\tag{30}
$$

which is, by definition, half the mean width of $H$ (the definition is the same for any convex body $K$). This quantity satisfies the hypotheses of Hadwiger's theorem, see Appendix. Furthermore, it has dimension of a length, so it is proportionnal to $V_1$. Hence for any convex body $K$, we have

$$
\int_{\mathbf{u} \in \mathbb{S}^{d-1}} d\nu(\mathbf{u}) \max_{\mathbf{r} \in K}(\mathbf{r}.\mathbf{u}) = \frac{V_1[K]}{V_1[\mathbb{B}^d]}.
\tag{31}
$$

The normalisation constant was found using the fact that for $K$ the unit ball $\mathbb{B}^d$, the max is attained for $\mathbf{r} = \mathbf{u}$ so the left-hand side equals 1. Using (A.7) we have

$$
V_1[\mathbb{B}^d] = \frac{d \, b_d}{b_{d-1}},
\tag{32}
$$

where recall $b_d = \mathrm{vol}(\mathbb{B}^d) = \pi^{d/2}/\Gamma(1+d/2)$. Putting everything together yields the first order expansion

$$\overline{\mathcal{V}_A}(\epsilon\,\mathbf{r}_1,\dots,\epsilon\,\mathbf{r}_{m-1}) = \epsilon\,\mathrm{vol}(\partial A)\frac{V_1[H]}{V_1[\mathbb{B}^d]} + O(\epsilon^2). \tag{33}$$

### 3.3.2 Second order contribution

Accessing the second order contribution is significantly more complicated. Let us go back to the expansion formula (25) for $\mathcal{V}_A$. Denote by $\partial A_j$ the regions in $\partial A$ for which $\max_k(0, \mathbf{r}_k.\mathbf{n}) = \mathbf{r}_j.\mathbf{n}$ and $\partial A_j^\epsilon$ the regions for which $\max_k\left(0, \mathbf{r}_k.\mathbf{n} + \epsilon\frac{\kappa_{ab}r_k^a r_k^b}{2}\right) = \mathbf{r}_j.\mathbf{n} + \epsilon\frac{\kappa_{ab}r_j^a r_j^b}{2}$. We may write [35]

$$\int_{\partial A} d\sigma \max_j\left(0, \mathbf{r}_j.\mathbf{n} + \epsilon\frac{\kappa_{ab}r_j^a r_j^b}{2}\right) = \sum_{j=1}^{m-1}\int_{\partial A_j^\epsilon} d\sigma\left[\mathbf{r}_j.\mathbf{n} + \epsilon\frac{\kappa_{ab}r_j^a r_j^b}{2}\right] \tag{34}$$

$$= \sum_{j=1}^{m-1}\int_{\partial A_j} d\sigma\left[\mathbf{r}_j.\mathbf{n} + \epsilon\frac{\kappa_{ab}r_j^a r_j^b}{2}\right] + o(\epsilon). \tag{35}$$

The last equation follows from the fact that $\partial A_j^\epsilon$ differs from $\partial A_j$ by a region of volume $\epsilon$, and an error of order $\epsilon$ is typically made in the integrand on such intervals. By typically, we mean that (35) holds for almost all vectors $\mathbf{r}_j \in \mathbb{R}^d$, but the formula might break at order $\epsilon$ if $\mathbf{r}_i.\mathbf{n} = \mathbf{r}_k.\mathbf{n}$ for some $i \neq k$. Said differently, for a given $\mathbf{n}$, (35) holds except for a set $(\mathbf{r}_1,\dots,\mathbf{r}_{m-1})$ of measure zero in $\mathbb{R}^d$. Since we shall shortly integrate over all possible orientations of $\mathbf{n}$ to obtain the average volume, we ignore this possibility altogether, as it does not affect the final result.

Let us now use (35) to simplify the average $\overline{\mathcal{V}_A}$. In particular, for each point $\mathbf{x} \in \partial A$, one can perform an average over the subgroup $SO(d-1)$ of rotations which leave the unit vector $\mathbf{n}$ invariant, and thus *do not affect* which index $j$ gets selected in the max function. Using

$$\int_{\mathcal{R}\in SO(d-1)} d\mu(\mathcal{R})\,\kappa_{ab}\mathcal{R}(r_j)^a\mathcal{R}(r_j)^b = \kappa_a^a\frac{(\mathbf{r}_j^\top)^2}{d-1} = (\mathrm{tr}\,\kappa)\frac{\mathbf{r}_j^2 - (\mathbf{r}_j.\mathbf{n})^2}{d-1}, \tag{36}$$

we may now undo all the steps performed to explicitly remove the max function, to obtain the alternative form

$$\overline{\mathcal{V}_A}(\mathbf{r}_1,\dots,\mathbf{r}_{m-1}) = \int_{\partial A} d\sigma \int_{\mathbf{u}\in\mathbb{S}^{d-1}} d\nu(\mathbf{u}) \max_{1\leq j\leq m-1}\left(0, \epsilon\,\mathbf{r}_j.\mathbf{u} + \epsilon^2\,\mathrm{tr}\,\kappa\frac{\mathbf{r}_j^2 - d(\mathbf{r}_j.\mathbf{u})^2}{2(d-1)}\right) + O(\epsilon^3), \tag{37}$$

where the average is taken on the unit sphere. We now replace the max over $j$ by a max over the Hull $H$. Similarly as before, the error made by such an approximation is of order $\epsilon^3$. The point behind such a replacement is that one can use Hadwiger's theorem, to obtain

$$\int_{\mathbf{u}\in\mathbb{S}^{d-1}} d\nu(\mathbf{u}) \max_{\mathbf{r}\in H}\left(\epsilon\,\mathbf{r}.\mathbf{u} + \epsilon^2\,\mathrm{tr}\,\kappa\frac{\mathbf{r}^2 - d(\mathbf{r}.\mathbf{u})^2}{2(d-1)}\right) = \epsilon\frac{V_1[H]}{V_1[\mathbb{B}^d]} - \frac{\epsilon^2}{2}(\mathrm{tr}\,\kappa)\frac{V_2[H]}{V_2[\mathbb{B}^d]} + O(\epsilon^3). \tag{38}$$

Indeed, one can check that the left-hand side of the previous equation is a valuation, and translation invariance up to order $\epsilon^3$ follows from the fact that the initial isotropic volume is translation invariant. To compute the coefficients on the right-hand side, the easiest is to compute them for a unit ball instead of the polytope $H$. Replacing the integrand in (37) by the previous equation yields

$$\overline{\mathcal{V}_A}(\epsilon\,\mathbf{r}_1,\dots,\epsilon\,\mathbf{r}_{m-1}) = \epsilon\,\mathrm{vol}(\partial A)\frac{V_1[H]}{V_1[\mathbb{B}^d]} - \frac{\epsilon^2}{2}\left(\int_{\partial A} d\sigma\,\mathrm{tr}\,\kappa\right)\frac{V_2[H]}{V_2[\mathbb{B}^d]} + O(\epsilon^3), \tag{39}$$

which is our main result. The intrinsic volumes can also be accessed by a more direct calculation. Introducing

$$\Omega_j = \left\{ \mathbf{u} \in \mathbb{S}^{d-1}, \mathbf{r}_j.\mathbf{u} = \max_{\mathbf{r} \in H}(\mathbf{r}.\mathbf{u}) \right\}, \tag{40}$$

we may rewrite the isotropic volume as

$$\overline{\mathcal{V}_A}(\epsilon\,\mathbf{r}_1,\ldots,\epsilon\,\mathbf{r}_{m-1}) = \epsilon\,\mathrm{vol}(\partial A)\,v_1[\mathbf{r}_1,\ldots,\mathbf{r}_{m-1}] - \epsilon^2\left(\int_{\partial A} d\sigma\,\mathrm{tr}\,\kappa\right)v_2[\mathbf{r}_1,\ldots,\mathbf{r}_{m-1}] + O(\epsilon^3), \tag{41}$$

where

$$v_1[\mathbf{r}_1,\ldots,\mathbf{r}_{m-1}] = \sum_{j=1}^{m-1}\int_{\mathbf{u}\in\Omega_j} d\nu(\mathbf{u})\,(\mathbf{r}_j.\mathbf{u}), \tag{42}$$

$$v_2[\mathbf{r}_1,\ldots,\mathbf{r}_{m-1}] = \sum_{j=1}^{m-1}\int_{\mathbf{u}\in\Omega_j} d\nu(\mathbf{u})\,\frac{d(\mathbf{r}_j.\mathbf{u})^2 - \mathbf{r}_j^2}{2(d-1)}. \tag{43}$$

One can check by repeated applications of Stokes' theorem that the resulting expression matches known results for the intrinsic volumes of polytopes [26]. This is discussed further below.

## 3.4  Discussion and special cases

Let us state the main technical result in a slightly different form. We have the large $\lambda$ expansion

$$\boxed{\begin{aligned} \overline{\mathcal{G}_{\lambda A}}(\mathbf{r}_1,\ldots,\mathbf{r}_{m-1}) = {} & \lambda^d\mathrm{vol}(A) - \alpha_d\lambda^{d-1}\mathrm{vol}(\partial A)V_1[H] \\ & + \beta_d\lambda^{d-2}\left(\int_{\partial A} d\sigma\,\mathrm{tr}\,\kappa\right)V_2[H] + o(\lambda^{d-2}), \end{aligned}} \tag{44}$$

where $\alpha_d = \frac{1}{d}\frac{b_{d-1}}{b_d}$ and $\beta_d = \frac{1}{2\pi(d-1)}$ are pure numbers which depend only on dimension ($b_d = \pi^{d/2}/\Gamma(1+d/2)$ is the volume of the unit ball). $\kappa$ denotes the curvature tensor at the boundary $\partial A$ and $\mathrm{tr}\,\kappa$ is its trace, the sum of all $(d-1)$ principal curvatures. $V_1$ and $V_2$ are the first and second *intrinsic volumes* of the convex Hull $H = H(\mathbf{r}_1,\ldots,\mathbf{r}_{m-1},0)$, which is the smallest convex polytope containing points $\mathbf{r}_1,\ldots\mathbf{r}_{m-1}, 0$, see Appendix (they are proportional to the $v_1$ and $v_2$ mentionned in Sec. 2).

Intriguingly, the above formula can be rewritten solely in terms of intrinsic volumes as

$$\begin{aligned} \overline{\mathcal{G}_{\lambda A}}(\mathbf{r}_1,\ldots,\mathbf{r}_{m-1}) = {} & \lambda^d\frac{V_0[\mathbb{B}^0]V_d[A]V_0[H]}{V_0[\mathbb{B}^d]} - \lambda^{d-1}\frac{V_1[\mathbb{B}^1]V_{d-1}[A]V_1[H]}{V_1[\mathbb{B}^d]} \\ & + \lambda^{d-2}\frac{V_2[\mathbb{B}^2]V_{d-2}[A]V_2[H]}{V_2[\mathbb{B}^d]} + o(\lambda^{d-2}), \end{aligned} \tag{45}$$

where $V_j[\mathbb{B}^d]$ are given by (A.7), and $V_0[\mathbb{B}^d] = 1$ was kept for aesthetic purposes. We emphasize that the appearance of such quantities is quite remarkable; we do not expect higher-order terms to have such a simple interpretation. Notice also that $A$ need not be convex for our main formula to hold, however a smooth boundary $\partial A$ is crucial.

Owing to locality, plugging (44) in (20) yields the cumulant expansion. Below we revisit special cases, expanding on what was mentionned in the introduction. We discuss the variance $m = 2$, the skewness $m = 3$, as well as the most usual dimensions $d = 2$ and $d = 3$, and finally the peculiar case of $d = 1$.

**The variance $m = 2$.** The convex Hull $H(\mathbf{r}, 0)$ is a line segment in any dimension. Since $H$ can be embedded in a one-dimensional space, $V_1[H] = |\mathbf{r}|$ coincides with the one-dimensional volume of the segment, and $V_2[H] = 0$. The expansion reads

$$\overline{\mathcal{G}_{\lambda A}}(\mathbf{r}) = \lambda^d \text{vol}(A) - \alpha_d \lambda^{d-1} \text{vol}(\partial A)|\mathbf{r}| + O(\lambda^{d-3}). \tag{46}$$

This particular result happens to be well-known in several different fields. For example it can be found in [39] in the context of small angle X-ray scattering, where recall $\overline{\mathcal{G}_{\lambda A}}(\mathbf{r})/\text{vol}(A)$ is called the (isotropic) correlation function. The isotropic covariogram $\overline{\mathcal{G}_{\lambda A}}(\mathbf{r}) = \overline{\mathcal{G}_{\lambda A}}(r)$ is also simply related to the distance distribution function, which is the probability density that two random points in $A$ are at distance $r$ (e.g., [25] and references therin), whose short distance asymptotics have been comprehensively studied.

**The skewness $m = 3$.** The convex Hull $H(\mathbf{r}_1, \mathbf{r}_2, 0)$ is a triangle in any dimension, so it can be embebbed in a two-dimensional plane. This means $V_1[H] = \frac{1}{2}\text{vol}(\partial H) = \frac{|\mathbf{r}_1| + |\mathbf{r}_2| + |\mathbf{r}_1 - \mathbf{r}_2|}{2}$, and $V_2[H] = \text{vol}(H) = \frac{|\mathbf{r}_1 \wedge \mathbf{r}_2|}{2}$, such that

$$\begin{aligned}
\overline{\mathcal{G}_{\lambda A}}(\mathbf{r}_1, \mathbf{r}_2) = \lambda^d \text{vol}(A) - \alpha_d \lambda^{d-1} \text{vol}(\partial A)\frac{|\mathbf{r}_1| + |\mathbf{r}_2| + |\mathbf{r}_1 - \mathbf{r}_2|}{2} \\
+ \beta_d \lambda^{d-2}\left(\int_{\partial A} d\sigma \, \text{tr}\,\kappa\right)\frac{|\mathbf{r}_1 \wedge \mathbf{r}_2|}{2} + O(\lambda^{d-3}).
\end{aligned} \tag{47}$$

**Two dimensions.** In two dimensions, the convex hull $H = H(\mathbf{r}_1, \ldots, \mathbf{r}_{m-1}, 0)$ is simply a convex polygon. Another simplification stems from the Gauss-Bonnet formula $\int_{\partial A} d\sigma \, \kappa = 2\pi \chi_A$ which relates mean curvature to the Euler characteristic ($\chi_A = 1$ for a simply connected region). The expansion reads

$$\overline{\mathcal{G}_{\lambda A}}(\mathbf{r}_1, \ldots, \mathbf{r}_{m-1}) = \lambda^2 \text{vol}(A) - \lambda \text{vol}(\partial A)\frac{\text{vol}(\partial H)}{2\pi} + \chi_A \text{vol}(H) + O(1/\lambda), \tag{48}$$

where $\text{vol}(\partial H)$ coincides with the perimeter of $H$. Notice that continuity of the intrinsic volumes imposes a convention where the perimeter is twice the length of $H$ if all vectors $\mathbf{r}_j$ are collinear.

**Three dimensions.** The convex hull $H = H(\mathbf{r}_1, \ldots, \mathbf{r}_{m-1}, 0)$ is a convex polyhedron, and the expansion reads

$$\overline{\mathcal{G}_{\lambda A}}(\mathbf{r}_1, \ldots, \mathbf{r}_{m-1}) = \lambda^3 \text{vol}(A) - \lambda^2 \text{vol}(\partial A)\sum_{e \in H}\frac{\ell_e \varphi_e}{8\pi} + \lambda\left(\int_{\partial A} d\sigma \, \text{tr}\,\kappa\right)\frac{\text{vol}(\partial H)}{8\pi} + O(1), \tag{49}$$

where the sum runs over all edges of the hull, $\ell_e$ is the length of the edge, and $\varphi_e$ is the corresponding (exterior) dihedral angle.

**One dimension.** By a direct calculation, one gets

$$\overline{\mathcal{G}_{\lambda A}}(x_1, \ldots, x_{m-1}) = \lambda \text{vol}(A) - \chi_A V_1[H], \tag{50}$$

where $\chi_A$ is the number of connected components, and

$$V_1[H] = \max_{1 \le j \le m-1}(0, x_j) - \min_{1 \le j \le m-1}(0, x_j), \tag{51}$$

is the volume (length) of the convex hull of $\{0, x_1, \ldots, x_{m-1}\}$. Formula (50) is consistent with the first two terms in (44) with convention $\alpha_1 = 1$, but it involves no approximation provided $\lambda$ is sufficiently large, so there are no further corrections.

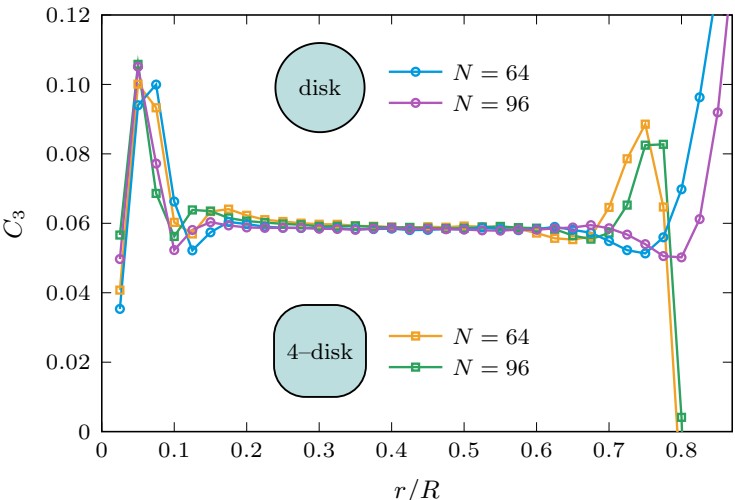

Figure 6: Third cumulant $C_3$ for a Laughlin state of bosons at filling fraction $\nu = 1/2$ in a droplet of radius $R$. Data for two regions with same Euler characteristic $\chi_A = 1$: the disk $x^2 + y^2 \leq r^2$ and a 4–disk deformation $x^4 + y^4 \leq r^4$. For large particle number $N$, $C_3$ becomes independent of the shape of region. The agreement deteriorates near $r = 0$ and $r = R$. This is due to boundary effects which break translational invariance for the latter, while in the former the region is not large enough for our results to apply. The relevant function, not known analytically, is $f(\mathbf{r}_1, \mathbf{r}_2) = \langle \rho(\mathbf{r}_1) \rho(\mathbf{r}_2) \rho(0) \rangle$ where $\rho$ is particle density.

## 4 An application: Quantum Hall states in 2d

We illustrate our general results on a concrete example, provided by 2d quantum Hall states of the form

$$\Psi(z_1, \ldots, z_N) = \prod_{1 \leq j < k \leq N} (z_j - z_k)^{1/\nu} e^{-\sum_{j=1}^{N} |z_j|^2/4}, \tag{52}$$

where $1/\nu$ takes integer values, and $z_k = x_k + i y_k$ are the coordinates of the particles. Here, $\nu$ is the filling fraction, and $\nu = 1$ corresponds to the integer quantum Hall effect (free fermions), $\nu = 1/2$ is the simplest interacting bosonic state, while $\nu = 1/3$ is the celebrated Laughlin $1/3$ state. For large particle number $N$, particles lie in a droplet of radius $R = \sqrt{2N/\nu}$ with constant density, while this same density goes to zero very quickly outside.

We look at particle fluctuations in large smooth regions in the bulk. Physically, all corresponding correlations become translationally invariant and isotropic when $N$ is large, which means our main formulas apply. Total particle number is also conserved. This implies the vanishing of $c_0, c_2$ for all even cumulants, and the vanishing of $c_0, c_1$ for all odd cumulants. Particle statistics in the state (52) are simulated with Monte Carlo techniques for arbitrary filling $\nu$. Most numerical results presented here are obtained using a procedure similar to that explained in [19, 40].

To demonstrate the validity of our main formula (13), in particular its topological nature, we consider the third cumulant or skewness. Since both the volume and area terms vanish, it is expected to converge to the constant

$$C_3(A) = \frac{\chi_A}{2} \int_{\mathbb{R}^4} d\mathbf{r}_1 d\mathbf{r}_2 |\mathbf{r}_1 \wedge \mathbf{r}_2| f(\mathbf{r}_1, \mathbf{r}_2), \tag{53}$$

for large $A$. To illustrate this result, we consider disks $x^2 + y^2 \leq r^2$ and deformed disks $x^4 + y^4 \leq r^4$ geometries, for various values of $r \in (0, R)$. Both regions share the same topology,

$\chi_A = 1$. Numerical results at filling fraction $\nu = 1/2$, see Fig. 6, confirm that for large regions and particle number the third cumulant becomes a constant, $C_3 \simeq 0.0583(1)$. We can compare this result with the one obtained by exploiting our relation (15) with the previously computed corner coefficient [20], $C_3 \approx 0.0597$. We can further generate a new prediction using the known [20] corner coefficient $\sigma_3$ for the Laughlin FQH state at $\nu = 1/3$, $C_3 \approx 0.049$. Another interesting example is provided by a sequence of annulus geometries for which $\chi_A = 0$. We have numerically checked that the skewness indeed vanishes asymptotically in that case.

We also performed several other checks at the free fermion point $\nu = 1$, for which Wick's theorem allows to reconstruct all correlations for the particle density. The integral in (53) can be evaluated analytically, yielding

$$C_3(A) = \frac{\chi_A}{2\sqrt{3}\pi}, \tag{54}$$

for arbitrary smooth region $A$. Expression (54) can be compared to exact computations of cumulants based on correlation matrix techniques (e.g., [31, 32, 41]), which typically allow to treat highly symmetric geometries only, such as the disk. We find precise agreement with our general formulas. We may further compute exactly the smooth limit of the corner term $\sigma_3 = 1/(4\sqrt{3}\pi^2) \simeq 0.0146245$, to be compared with the numerical calculation $\sigma_3 \simeq 0.01462$ performed in [19], both in perfect agreement.

## 5    Discussion

We have developed a method for computing arbitrary cumulants of local observables in arbitrary smooth regions for arbitrary translation invariant theories, under physically natural assumptions on locality.

We have established a general explicit formula (3–5) for the first orders in the expansion (2) of the cumulants for large regions, showing a complete factorisation between the geometric and physics contributions. The latter is encoded in what we dubbed "convex moments" of the connected correlation functions (6–8), which are integrals of the connected $m$–point functions against certain *intrinsic volumes* of convex hulls of $m$ points. An intrinsic volume can be interpreted as the average shadow of a convex body projected onto a plane (see Fig. 1 (c)). The appearance of such volumes in our problem is very surprising, since the only geometric input is the choice of a smooth non-necessarily convex region $A$.

Subleading to the volume and area terms, the second-order coefficient in the cumulants' expansion is sensitive to the integrated boundary curvature, which can be related in two dimensions to the Euler characteristic of the region $A$—a topological invariant. Remarkably, odd cumulants of conserved observables for which the volume and area terms vanish are thus topological in two dimensions, see (13). Said differently, odd cumulants remain unaffected by smooth deformations of the (large) region's shape. To illustrate this particular feature of odd cumulants and to demonstrate the validity of our general results, we have performed Monte Carlo calculations of the third cumulant for different shapes of region in a strongly-interacting system provided by the two-dimensional quantum Hall state at filling fraction $\nu = 1/2$. The numerical results perfectly agree with our theoretical prediction.

To derive our results, we have assumed a fast decay of correlations, which is guaranteed, e.g., in gapped phases with a finite correlation length. Such an assumption was mainly technical, the generality of our results should extend well beyond gapped phases: a rule of thumb is that the asymptotic expansion remains valid up to a finite order, specifically up to the point where the convex moments $s_j[f]$ cease to be finite. This typically happens in theories where correlations decay algebraically (i.e., as a power law), in which case the asymptotic expansion

holds as long as the decay exponent is large enough to ensure convergence of the relevant integrals or sums defining the moments. A particularly interesting case occurs when the convex moments have a mild logarithmic divergence. Several physical examples have been already discussed for the variance [15,19], in which case the geometric part is enhanced by a multiplicative $\log \lambda$ term, and the remaining physical part can be universal. We expect a similar phenomenon to occur for higher cumulants.

An important question regards the physical content behind such convex moments. In the simple case of the variance in two dimensions, it was shown [19] that, for geometries with corners, the corresponding corner terms are related to the (universal) small momentum expansion of the structure factor, related to the Stillinger-Lovett sum rule. We do not know whether our new convex moments can, in some cases, be connected to universal higher sum rules of the correlation functions.

We have focused on the first three orders in the expansion of cumulants at large region size, which already present rich features. In a follow-up paper, we discuss higher orders and the more complicated method allowing their derivation. Furthermore, it would be interesting to revisit our derivation of cumulants' expansion assuming less symmetry, for example without rotational invariance for anisotropic systems, or in curved space.

## Acknowledgments

We are grateful to G. Aubrun for pointing out the relation between some of our formulas and the concept of intrinsic volumes in convex geometry.

**Funding information** W.W.-K. is supported by a grant from the Fondation Courtois, a Chair of the Institut Courtois, a Discovery Grant from NSERC, and a Canada Research Chair.

## A  Appendix

Intrinsic volumes are fundamental functionals in stochastic and integral geometry, also known as geometric probability [25,26]. For instance, they appear in the probability that one body moving uniformly at random will intersect with another body, via the principal kinematic formula. Intrinsic volumes provide a complete set of measurements that characterize the size and structure of a convex body, or more generally, a polyconvex set (a finite union of convex bodies). They feature in the volume of the $\epsilon$-neighborhood of a compact convex subset $K$ of $\mathbb{R}^d$ (the set of points within a distance $\epsilon$ from $K$), as given by the Steiner formula:

$$\mathrm{vol}_d\big(K + \epsilon\,\mathbb{B}^d\big) = \sum_{j=0}^{d} b_{d-j}\epsilon^{d-j}V_j[K]. \tag{A.1}$$

Here, $\mathrm{vol}_d$ stands for the $d$-dimensional volume, $\mathbb{B}^d$ denotes the unit ball in $\mathbb{R}^d$, and $b_j = \mathrm{vol}_j(\mathbb{B}^j) = \pi^{j/2}/\Gamma(j/2+1)$. The functional $V_j$ is called the $j$-th intrinsic volume. Up to normalization, $V_j[K]$ is the average volume of the projection of $K$ onto a $j$-dimensional subspace of $\mathbb{R}^d$, chosen uniformly at random (Kubota's formula):

$$V_j[K] = \binom{d}{j}\frac{b_d}{b_j b_{d-j}}\int_{G(d,j)} d\,\nu_j(L)\,\mathrm{vol}_j(K|L), \tag{A.2}$$

where $G(d,j)$ is the Grassmannian of $j$-dimensional linear subspaces of $\mathbb{R}^d$, equipped with the

invariant probability measure $\nu_j$, and $\mathrm{vol}_j(K|L)$ is the $j$-dimensional volume of the orthogonal projection of $K$ onto $L$.

Intrinsic volumes capture essential geometric properties such as volume, surface area, and mean width. Specifically, $V_d[K] = \mathrm{vol}_d[K]$ is the ($d$-dimensional) volume of $K$, while $2V_{d-1}[K] = \mathrm{vol}_{d-1}(\partial K)$ is the surface area of $K$. Up to prefactors, $V_{d-2}[K]$ is the mean curvature of $\partial K$, and $V_1[K]$ is the mean width of $K$. Finally for $K$ a non-empty compact convex set, $V_0[K] = 1$, and more generally for a polyconvex set $V_0[K] = \chi(K)$ gives the Euler characteristic. If the boundary $\partial K$ is a smooth hypersurface, all intrinsic volumes (except for $V_d$) can be expressed as integrals of local geometric quantities on the surface $\partial K$. Specifically,

$$V_j[K] = \frac{1}{(d-j)b_{d-j}} \int_{\partial K} d\sigma \, e_{d-j-1}(\kappa_1, \cdots, \kappa_{d-1}), \tag{A.3}$$

where $e_p$ are the elementary symmetric functions in the principal curvatures $\kappa_i$,

$$e_p(\kappa_1, \ldots, \kappa_{d-1}) = \sum_{1 \le i_1 < \ldots < i_p \le d-1} \kappa_{i_1} \ldots \kappa_{i_p}, \tag{A.4}$$

with the convention $e_0 = 1$. These curvature integrals allow to extend the notion of intrinsic volumes to any compact domain $A$ with a smooth boundary $\partial A$. In particular,

$$V_0[A] = \frac{1}{d b_d} \int_{\partial K} d\sigma \prod_{j=1}^{d-1} \kappa_j = \chi_A, \tag{A.5}$$

is the Euler characteristic.

The terminology *intrinsic* volumes comes from the fact they do not depend on the dimension into which the convex subset is embedded. Besides this property, intrinsic volumes exhibit many important characteristics:

- They are non-negative ($V_j[K] \ge 0$) and monotone ($K_1 \subset K_2$ implies $V_j[K_1] \le V_j[K_2]$).

- They are homogeneous of degree $j$: $V_j[\lambda K] = \lambda^j V_j[K]$.

- They depend continuously on $K$.

- They are *valuations* on convex bodies (*i.e.* compact convex subsets with non-empty interior). Roughly speaking a valuation is a *notion of size*. More precisely this means that $V_j[\emptyset] = 0$ and that given two convex bodies $K_1$ and $K_2$ such that $K_1 \cup K_2$ is also a convex body, then

$$V_j[K_1 \cup K_2] = V_j[K_1] + V_j[K_2] - V_j[K_1 \cap K_2]. \tag{A.6}$$

  This additivity property allows to extend the notion of intrinsic volumes to finite unions of convex bodies.

- They are invariant under all isometries of $\mathbb{R}^d$.

Hadwiger's theorem ensures that the above properties characterize uniquely intrinsic volumes up to normalization. More precisely, any invariant, continuous valuation on convex bodies in $\mathbb{R}^d$ is a linear combination of the intrinsic volumes $V_0, V_1, \ldots, V_d$. In particular, any invariant, continuous valuation that is homogeneous of degree $j$ is proportional to $V_j$. This profound theorem has intriguing applications. For example, in three dimensions, the average projected area of a convex solid is one-quarter of its surface area, a result proven by Cauchy in the 19th century. There are also applications in grain growth theory [30], among others.

In the following, we will also need the intrinsic volumes of the unit sphere $\mathbb{B}^d$. Those can simply be obtained from either Steiner (A.1) or Kubota's formula (A.2) as

$$V_j[\mathbb{B}^d] = \binom{d}{j} \frac{b_d}{b_{d-j}}. \tag{A.7}$$

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
