# Peer review of "Geometric expansion of fluctuations and average shadows"

_SciPost Physics, doi:SciPost Phys. 19, 122 (2025)_

## Round 2 · Referee Report · Anonymous (Referee 1) · 2025-8-15

Strengths

  • Solid analytical results.
  • Coherent links to other quantities and fields.
  • The individual results are clearly stated.

Weaknesses

  • The global structure of the paper is a very meandering and would be helped by a minor refactoring.
  • By structure, it focuses on gapped phases (or at least on terms which are not affected by the gaplessness of the Hamiltonian). Anomalous contributions to correlators in gapless systems have seen a lot of interest.

Report

This paper presents a generic derivation of the large domain expansion of the integrated connected correlation functions / fluctuations of some observables.
The derivation is well-explained, and universal, thanks to a clear factorization between physical observable and geometrical factors. Crucially, it only relies on global rotation and translation invariance, and therefore also applies to strongly interacting systems.

The results are solid and provide interesting directions to study a wide variety of models thanks to the explicit formula derived in this paper. As such I would recommend publication in Scipost Physics after minor revisions.

Indeed, I did find the paper sometimes hard to follow in its logic. While each point is in itself quite understandable, the flow keeps going back and forth. I would strongly recommend a refactoring of the paper.
As an example: Eq. 9 and Eq. 11 appear to differ by a factor of 2. It is only below Eq. 29 that the counting convention is made explicit.

Additionally, there is not really any discussions of the logarithmic corrections that can arise in gapless models, despite the significant interest they raise (as the authors very well know). While I understand the generalization to this type of models might be out of scope for this paper, at least a discussion of what and where the approach fails, as well as some general connecting comments would be appreciated.

Requested changes

1) Refactoring the structure of the paper would be a plus, to streamline the presentation and avoid some repetitions (typically results for variance and skewness are mentioned several times throughout the paper).

2) There is no connection to the anomalous logarithmic corrections, which I would appreciate.

3) While the geometrical contributions are extremely well described, there are no discussions of the physical meaning of the observable part.

Minor change: - "letter" should be changed in the introduction

Recommendation

Ask for minor revision

---

## Round 2 · Referee Report · Anonymous (Referee 2) · 2025-8-28

Strengths

The paper presents a clear derivation of the results which are new and interesting to a large community

Weaknesses

The structure could be improved slightly: - the convention to compute volumes below Eq. (29) needs to be given explicitly earlier in the paper - in section III maybe the flow would be more natural by first deriving the formulae and then giving and analysing the formula including the special cases (there is indeed already a section for the main results)

Report

The paper reports on the interesting question of full counting statistics which arise naturally in condensed matter and statistical mechanics. The authors derive the general expression of the cumulants of the number of particles in a domain A as the size of the domain becomes increasingly large and under simple assumptions of homogeneity, isotropy and sufficiently fast decay of correlations.
They derive expressions for the cumulants that clearly separate the geometric contribution (which are described in details) to the physical contributions.

The paper derives new and interesting results that will benefit the community working on these topics. I therefore recommend the paper for publication in Scipost Physics after some minor changes.

Requested changes

See suggestions in "weaknesses"

Recommendation

Ask for minor revision

---

## Round 3 · Author Response

We thank both referees for their thorough analysis of our paper, their positive reviews and the points they raised.

As suggested by the referees, we did a refactoring of our paper (in particular section III) for a streamlined presentation and a better flow.
We now discuss how the generality of our results extend beyond gapped phases, as well as what we know of the physical content behind the convex moments.

---

## Round 3 · List of Changes

• Structure improved (Section III in particular)
  • Discussion expanded
  • The convention to compute volumes that was given below Eq. (29) is now given explicitly below Eq. (12)

---

## Editorial Decision

published